# Direct Chromosomal Phasing: An Easy and Fast Approach for Broadening Prenatal Diagnostic Applicability

**Stefania Byrou [1], George Christopoulos [1], Agathoklis Christofides [2], Christiana Makariou [3], Christiana Ioannou [3], Marina Kleanthous [1,†] and Thessalia Papasavva [1,*,†]**

1 Molecular Genetics Thalassaemia Department, The Cyprus Institute of Neurology and Genetics, Nicosia 2371, Cyprus; stefaniab@cing.ac.cy (S.B.); gchrist@cing.ac.cy (G.C.); marinakl@cing.ac.cy (M.K.)

2 Medical Centre Fetal Medicine Department, Archbishop Makarios III Hospital, Nicosia 2371, Cyprus; agathoklis_christofides@yahoo.gr

3 Thalassaemia Screening Laboratory, Thalassaemia Center, Archbishop Makarios III Hospital, Nicosia 2371, Cyprus; chmakariou@gmail.com (C.M.); athens2612@gmail.com (C.I.)

* Correspondence: thesalia@cing.ac.cy; Tel.: +357-22392664

† MK and TP are Joint Senior Authors.

**Abstract:** The assignment of alleles to haplotypes in prenatal diagnostic assays has traditionally depended on family study analyses. However, this prevents the wide application of prenatal diagnosis based on haplotype analysis, especially in countries with dispersed populations. Here, we present an easy and fast approach using Droplet Digital PCR for the direct determination of haplotype blocks, overcoming the necessity for acquiring other family members' genetic samples. We demonstrate this approach on nine families that were referred to our center for a prenatal diagnosis of β-thalassaemia using four highly polymorphic single nucleotide variations and the most common pathogenic β-thalassaemia variation in our population. Our approach resulted in the successful direct chromosomal phasing and haplotyping for all nine of the families analyzed, demonstrating a complete agreement with the haplotypes that are ascertained based on family trios. The clinical utility of this approach is envisaged to open the application of prenatal diagnosis for β-thalassaemia to all cases, while simultaneously providing a model for extending the prenatal diagnostic application of other monogenic diseases as well.

**Keywords:** haplotyping; chromosomal phasing; prenatal diagnosis; β-thalassaemia; droplet digital PCR; SNVs



## 1. Introduction

Traditional DNA sequencing strategies are often limited to the determination of the nucleotide content of the targeted loci and do not segregate or 'phase' the sequencing information obtained from maternally and paternally-inherited chromosomes. However, many studies support that the assignment of variant alleles to each of the two homologous copies of a gene is essential for understanding the genotype–phenotype correlation and disease manifestation [1,2].

Phase information is important for a variety of molecular biology scenarios, including allele-specific expression [3,4], compound heterozygosity [5], and prenatal diagnosis [6,7]. In allele-specific expression studies, the altered rate of expression of heterozygous cis-acting variants compared to the normal allele affects gene function and in turn protein expression; therefore, knowledge of the haplotypic arrangements of variants on each homologue would be of significant value [8]. In addition, the determination of the position of deleterious variants on the same copy of a gene (*in-cis configuration*) or on opposite copies (*in-trans configuration*) is important, as the *in-trans* configuration in cases of compound heterozygosity potentially inactivates both copies, thus leading to reduced gene function [9]. Importantly, leveraging information from parents or other family members for chromosomal phasing

can lead to the identification of disease-predisposing variants in the offspring providing prenatal diagnosis in a highly-accurate manner [1].

Prenatal diagnosis approaches are commonly confirmed using two methods: the direct and indirect mutation detection methods. Haplotype analysis provides an indirect prenatal diagnostic method that serves many purposes, such as the determination of the phase of the inherited allele in the fetus, i.e., disease-causing or wild-type, the exclusion of maternal tissue contamination in the fetal sample and sample mix-up, and the marking of recombination events during meiosis [10]. In that way, the reliability of the prenatal test is being increased, simultaneously offering higher diagnostic power and accuracy. Moreover, the risk for misdiagnosis is remarkably reduced when the direct mutation detection method is coupled with haplotype analysis, since, in case of discordant results after the comparison of the two independent methods, the test will be repeated, and therefore a misdiagnosis is prevented.

As the gold standard, haplotype analysis approaches are based on the requirement for family members to provide their genetic material, such as homozygote wild-type or homozygote mutant probands or others [11], to be able to link the inferred haplotypes with the disease under investigation. Family-based data provide useful information for resolving the allelic phase and constructing the haplotypes of the target individual. However, the availability of the genetic material from a proband is usually stringent for several reasons, such as a possible death or long distance.

The routine performance of haplotype analysis for the prenatal diagnosis of β-thalassaemia in our center pinpointed the challenge that is faced when family members are not available for prenatal diagnostic applications and the importance of developing an assay that bypasses this prerequisite.

β-thalassaemia belongs to the broad category of β-haemoglobinopathies, a group of prevalent, autosomal recessive single-gene disorders of the blood, which are potentially fatal if untreated. β-thalassaemia is exceptionally prevalent in Cyprus, with an approximate 12% carrier frequency, and with the HBB$^{IVSI-110\,G>A}$ (HGVS ID HBB:c.93-21G>A) β-globin pathogenic variant representing 79.8% of the total [12]. Due to the high prevalence of the disease in Cyprus, a thalassaemia prevention program has been operating in the island since 1981 [13], which includes measures for informing and educating the public, the premarital screening of the population, genetic counseling, and prenatal diagnosis.

The development of a direct chromosomal phasing method that can connect an individual's haplotypes with the disease directly and bypass the need for other family members' genetic material is of paramount importance for broadening the prenatal test's applicability to more families.

Recently, direct chromosomal phasing approaches have been presented, seeking to determine the phase of neighboring alleles using information gained from a single individual. In 2014, the Targeted Locus Amplification (TLA) strategy was presented [14]. This approach requires cells as an input to selectively amplify and sequence whole genes. The TLA technology enables many applications, such as the phasing of single blocks of heterozygous variants and the detection of Single Nucleotide Variants (SNVs), gene fusions, and structural variants. In 2016, the group of Zheng et al. developed a linked-read technology based on microfluidics and proved its efficacy on phasing and haplotyping cancer and germline genomes [15]. Despite their high efficacy, both TLA and linked-read technologies are fastidious and expensive, requiring a great deal of experimental procedures and specialized equipment.

In our group's effort to develop a robust and simple method for direct chromosomal phasing in family studies for the prenatal diagnosis of β-thalassaemia, the Drop-phase [16] assay was assessed. This approach allows for the determination of the phase of DNA variant pairs using Droplet Digital PCR (ddPCR), even at genomic distances of 200 Kb, relying on the idea that "when two alleles are physically linked, they tend to partition into the same droplets" [16]. Exploiting the single molecule analysis potentials of ddPCR,

chromosomal phasing provides information on the inheritance pattern of specific alleles when duplexed with others.

Herein, we present the development of a direct chromosomal phasing assay using five SNVs located on the β-globin gene cluster including one pathogenic that is flanked by the SNVs, and advocate for the introduction and optimization of more SNVs than those currently used. The introduction of a direct chromosomal phasing assay into a genetics laboratory would be significant for determining the inheritance patterns within families without using generational information.

Below, we describe an easily-adopted, time- and cost-efficient method for the construction and assembly of haplotype blocks in β-thalassaemia prenatal diagnostics by solely genotyping the parental DNA samples and overcoming the dependence on relatives' sample availability.

## 2. Materials and Methods

### 2.1. Ethics Statement

The study was approved by the Cyprus National Bioethics Committee (EEBK/EΠ/2018/51). All participants gave informed written consent (EEBK03) in accordance with local laws and regulations. All procedures were conducted in accordance with the Declaration of Helsinki, ensuring the confidentiality of sample-related data. The current study is part of a wider non-invasive prenatal testing (NIPT) project for the development of an integrated Non-Invasive Prenatal Haplotyping assay for β-thalassaemia. The current assay of direct parental chromosomal phasing will work as tool to facilitate a more efficient, integrated NIPT to be implemented in the clinical setting.

### 2.2. Sample Collection

Peripheral blood samples and the corresponding chorionic villi samples (CVS) were collected from 9 family trios (mother, father, and CVS) for the assessment of the chromosomal phasing project. The families were referred to our laboratory for the routine invasive Prenatal Diagnosis analysis and gave their informed consent for participation in the ongoing NIPT study. Approximately 3 mL of peripheral blood samples were collected into EDTA-containing tubes from the 9 couples and 1 mL was used for genomic DNA isolation using the Gentra® Puregene® Kit (Qiagen, Hilden, Germany). The CVS sample was collected from pregnant women, between their 11th and 12th week of gestation, in saline solution (0.9% NaCl), and fetal genomic DNA was extracted using the Nucleospin® Tissue Kit (Macherey-Nagel, Düren, Germany).

### 2.3. Single Nucleotide Variant Selection and Genotyping

The selection of SNVs to be included in this assay was based on the pre-determined SNP panel of Papasavva et al., [17] to obtain high heterozygosity SNPs located on the β-globin gene cluster. Five variants, including one pathogenic variant, were selected [SNV rs1003586 (G/A), SNV rs7480526 (G/T), SNV rs11036364 (T/C), SNV rs2187610 (G/C) and SNV IVSI-110 (c.93-21G>A)]. The maximum distance between the 5 variants was 3964 bp and the minimum was 366 bp. To prevent false results during haplotyping in case of recombination events, SNVs were selected such that they were located in flanking positions with respect to the pathogenic SNV (Figure 1). All genomic DNA samples used in the study were previously genotyped for the selected variants using MALDI-TOF MS (Sequenom® GmbH, Hamburg, Germany) [17].

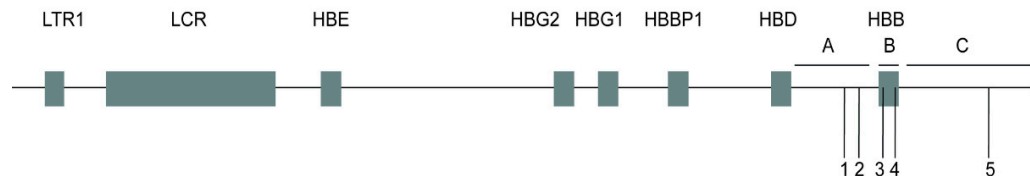

**Figure 1.** Schematic representation of the variant location on the β-globin cluster analyzed for direct chromosomal phasing. LTR1: Long Terminal Repeat, LCR: Locus Control Region, HBE: Haemoglobin Subunit Epsilon, HBG2: Haemoglobin Subunit Gamma-2, HBG1: Haemoglobin Subunit Gamma-1, HBBP1: Haemoglobin Subunit Beta Pseudogene 1, HBD: Haemoglobin Subunit Delta, HBB: Haemoglobin Subunit Beta, A: δ–β intergenic region, B: β intragenic region and C: post-β distal region, 1: SNV rs1003586 (G/A), 2: SNV rs11036364 (T/C), 3: pathogenic SNV IVSI-110 (c.93-21G>A), 4: SNV rs7480526 (G/T), and 5: SNV rs2187610 (G/C).

## 2.4. Oligonucleotide Design

Primers (Table 1) and allele-specific FAM and HEX fluorescently-labeled probes (Table 2) with Black Hole Quencher® (BHQ-1) were designed specifically for each SNV.

**Table 1.** Primer sequences.

| Variant | Primer ‖ F (5′-3′) | Primer ¶ R (5′-3′) | Amplicon Size |
|---|---|---|---|
| rs1003586 (G/A) | GGAAAAAGTACAGGGGGAT | TTATATTTTGTTTGTTTAAACCTCCTT | 101 bp |
| rs11036364 (T/C) | GTATTTTTGACTGCATTAAGAGG | GCTCTGTGCATTAGTTACTTAT | 80 bp |
| IVSI-110 (c.93-21G>A) | GCATGTGGAGACAGAGAA | ACCACCAGCAGCCTAA | 93 bp |
| rs7480526 (G/T) | TCCCCTTCTTTTCTATGGTTAA | GATGCAATCATTCGTCTGTT | 90 bp |
| rs2187610 (G/C) | AGTGAGGCTACATCAAACTAA | CCATTTCATGGTTCACCTTT | 78 bp |

‖ Forward, ¶ Reverse.

**Table 2.** Allele-specific, fluorescently-labeled oligonucleotide sequences.

| Variant | Probe FAM | Probe HEX |
|---|---|---|
| rs1003586 [G/A] | ATCACGTTGGGAAGCTATAGAGA | TCACGTTGGAAAGCTATAGAGAAAG |
| rs11036364 [T/C] | TCTAGTTTTTTATCTCTTGTTTCCCA | AGTTTTTTACCTCTTGTTTCCCAAAAC |
| IVSI-110 (c.93-21G>A) | TCTCTGCCTATTGGTCTATTTTCC | TCTGCCTATTAGTCTATTTTCCCAC |
| rs7480526 [G/T] | ATGTCATAGGAAGGGGAGAAGTAA | TTCATGTCATAGGAAGGGGATAAGTA |
| rs2187610 [G/C] | TCCACACAAAAAAGAAAACAATGAA | TTTCCACACAAAAAACAAAACAATGA |

To eliminate the cross-reactivity of the fluorescently-labeled probes with the non-targeted allele and further enhance the robustness of the assay, non-fluorescent competitor probes (Table 3) were also included in the assay [16]. These probes are complementary to the non-targeted allele, and upon binding, prevent the hybridization of the fluorescently-labeled probe [16]. The non-fluorescent competitor internal oligos do not have a reporter dye at their 5′ end because their role is to enhance the effect of competing for the non-specific binding site based on their full complementarity to the non-targeted allele. All oligonucleotides were designed based on the GRCh37 assembly, with the use of the Primer3plus software, 3.2.6, Andreas Untergasser and Harm Nijveen, Boston, MA, USA.

**Table 3.** Non-fluorescent, competitor oligonucleotide sequences.

| Variant | Competitor Oligo for FAM | Competitor Oligo for HEX |
|---|---|---|
| rs1003586 | TCACGTTGGAAAGCTATAGAGAAAG | ATCACGTTGGGAAGCTATAGAGA |
| rs11036364 | AGTTTTTTACCTCTTGTTTCCCAAAAC | TCTAGTTTTTTATCTCTTGTTTCCCA |
| IVSI-110 (c.93-21G>A) | TCTGCCTATTAGTCTATTTTCCCAC | TCTCTGCCTATTGGTCTATTTTCC |
| rs7480526 | TTCATGTCATAGGAAGGGGATAAGTA | ATGTCATAGGAAGGGGAGAAGTAA |
| rs2187610 | TTTCCACACAAAAAACAAAACAATGA | TCCACACAAAAAAGAAAACAATGAA |

## 2.5. Drop-Phase Assay Principle

While relying on the ddPCR efficiency for single molecule analysis, the Drop-phase assay can provide information on the inheritance motif of specific alleles when duplexed. Primers and fluorescently-labelled probes that target two SNV alleles at different loci are duplexed and processed for droplet generation. A PCR reaction is then performed within each droplet and the droplets are processed for analysis on the Droplet Reader. Since there is a single molecule per droplet and due to the co-existence of physically-linked alleles on the same chromosome (*cis-configured alleles*), they are also expected to co-partition in the same droplets, thus emitting a double-positive signal. On the other hand, when the two SNV alleles under investigation are located on different chromosomes (*trans-configured alleles*), they will not co-partition into the same droplet, thus exhibiting little or no double-positive droplets. Hence, *cis-configured alleles* will result in a much greater number of double-positive droplets than the trans-configured alleles, allowing for the deduction of the phase of each allele. The FAM-positive droplets emit a blue signal and the HEX-positive a green signal. Double-positive droplets result in an orange population of droplets on the Digital PCR results plot while the double-negatives result in a grey population (Figure 2).

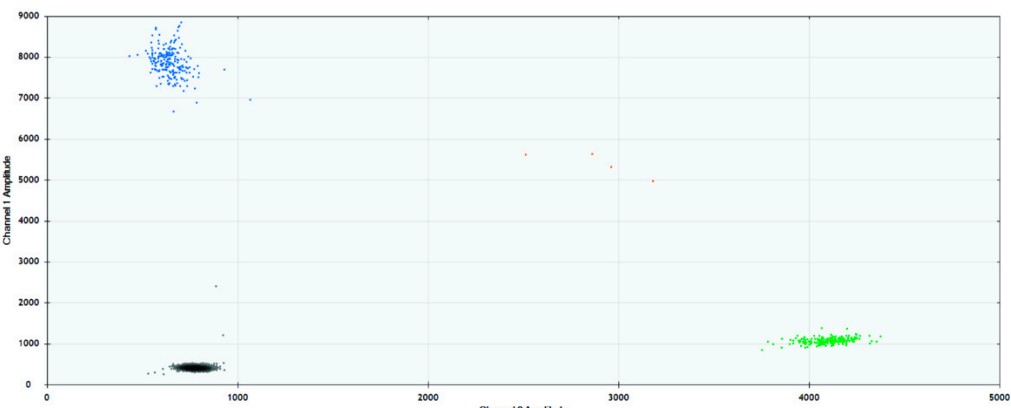

**Figure 2.** 2D-plot of droplet fluorescence. Blue color demonstrates the FAM-positive droplets while the green presents the HEX-positive droplets. The orange signal shows the double-positive events on the Digital PCR results plot and the grey color represents the double-negative droplet population.

Ten multiplex reactions were performed for each family member, each targeting two SNVs. For each reaction, multiplexing of the two primer pairs with the corresponding probes that target a different SNV (one FAM-labeled and one HEX-labeled) and two competitor oligonucleotides (one for each probe) was performed, as described in Table 4. Duplexing of the two probes in a reversed manner was also performed to cross-confirm the results, as shown in Figure 3. Each of the 5 SNVs was duplexed with 2 other SNVs (Figure 3).

**Table 4.** Duplexed reaction information for each sample.

| Duplex | Primers | Probes | Competitor Oligo |
|---|---|---|---|
| 1 | rs1003586 ‖ F & ¶ R | rs1003586_G_FAM | rs1003586_competitor for G († CAM) |
| | rs11036364 F & R | rs11036364_C_HEX | rs11036364_competitor for C (‡ CEX) |
| 2 | rs1003586 F & R | rs1003586_A_HEX | rs1003586_competitor for A (CEX) |
| | rs11036364 F & R | rs11036364_T_FAM | rs11036364_competitor for T (CAM) |
| 3 | rs11036364 F & R | rs11036364_T_FAM | rs11036364_competitor for T (CAM) |
| | IVSI-110 F & R | IVSI-110_A_HEX | IVSI-110_competitor for A (CEX) |
| 4 | rs11036364 F & R | rs11036364_C_HEX | rs11036364_competitor for C (CEX) |
| | IVSI-110 F & R | IVSI-110_G_FAM | IVSI-110_competitor for G (CAM) |
| 5 | IVSI-110 F & R | IVSI-110_G_FAM | IVSI-110_competitor for G (CAM) |
| | rs7480526 F & R | rs7480526_T_HEX | rs7480526_competitor for T (CEX) |
| 6 | IVSI-110 F & R | IVSI-110_A_HEX | IVSI-110_competitor for A (CEX) |
| | rs7480526 F & R | rs7480526_G_FAM | rs7480526_competitor for G (CAM) |
| 7 | rs7480526 F & R | rs7480526_G_FAM | rs7480526_competitor for G (CAM) |
| | rs2187610 F & R | rs2187610_C_HEX | rs2187610_competitor for C (CEX) |
| 8 | rs7480526 F & R | rs7480526_T_HEX | rs7480526_competitor for T (CEX) |
| | rs2187610 F & R | rs2187610_G_FAM | rs2187610_competitor for G (CAM) |
| 9 | rs2187610 F & R | rs2187610_G_FAM | rs2187610_competitor for G (CAM) |
| | rs1003586 F & R | rs1003586_A_HEX | rs100358610_competitor for A (CEX) |
| 10 | rs2187610 F & R | rs2187610_C_HEX | rs2187610_competitor for C (CEX) |
| | rs1003586 F & R | rs1003586_G_FAM | rs100358610_competitor for G(CAM) |

‖ Forward, ¶ Reverse, † Competitor for FAM, ‡ Competitor for HEX.

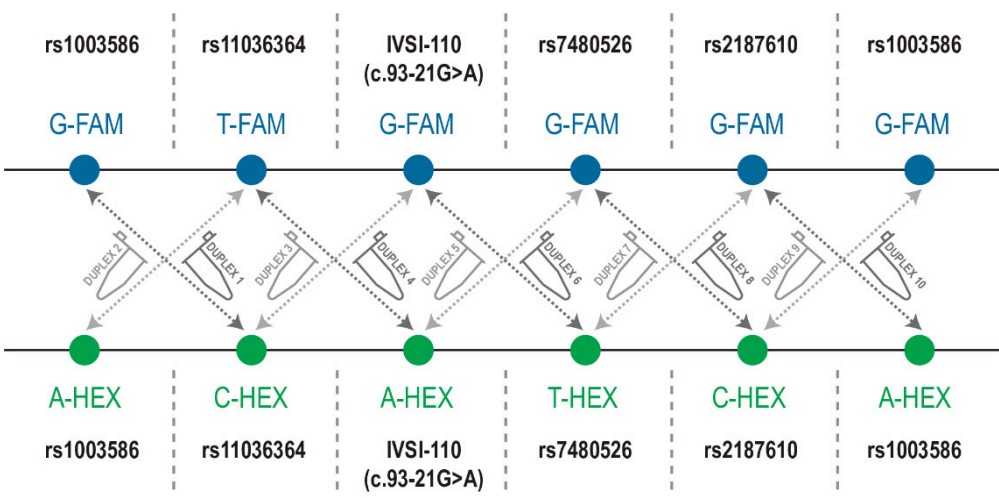

**Figure 3.** Schematic representation of duplexing for each sample.

Variant alleles colored in blue represent the FAM probe whereas the alleles colored in green represent the HEX probe. The arrows demonstrate which two probes are being duplexed in each reaction. In the first duplex, the G allele of SNV rs1003586 that is FAM-labeled, is multiplexed with the HEX-labeled probe of SNV rs11036364 that targets the C allele. For cross-confirmation, in a second duplex reaction, the HEX probe that targets the A allele of SNV rs1003586 is multiplexed with the FAM probe that targets the T allele of SNV rs11036364. Next, in the third duplex, the same probe is multiplexed with the HEX probe of the pathogenic variant IVSI-110 (c.93-21G>A) that targets the A allele, and

in the fourth duplex, the FAM probe that targets the G allele of the pathogenic variant IVSI-110 (c.93-21G>A) is multiplexed with the HEX probe of SNV rs11036364 that targets the C allele. In the fifth duplex, the FAM probe that targets the G allele of the pathogenic variant IVSI-110 (c.93-21G>A) is multiplexed with the HEX probe of SNV rs7480526 that targets the T allele, while the sixth duplex consists of the HEX probe that targets the A allele of the pathogenic variant IVSI-110 (c.93-21G>A) and the FAM probe of SNV rs7480526 that targets the G allele. Moreover, the same probe is multiplexed in the seventh duplex with the HEX probe that targets the C allele of SNV rs2187610, while in the eighth duplex, the HEX probe that targets the T allele of SNV rs7480526 is multiplexed with the FAM probe that targets the G allele of SNV rs2187610. The ninth duplex consists of the FAM probe that targets the G allele of SNV rs2187610 and the HEX probe of SNV rs1003586 that targets the A allele. In the tenth and final duplex, the HEX probe that targets the C allele of SNV rs2187610 is multiplexed with the FAM probe that targets the G allele of SNV rs1003586.

*2.6. Droplet Digital PCR*

A total of 20 ng of genomic DNA was used as a template in a reaction volume of 25 μL that contained 1× ddPCR™ Supermix for Probes (no dUTP) (Bio-Rad, Hercules, CA, USA), 450 nM for each set of primers, 250 nM of each fluorescently-labeled probe, and 1000 nM of each competitor oligo. A total of 20 μL of the reaction mix was added to DG8™ Cartridges for QX200™ Droplet Generator (Bio-Rad, Hercules, CA, USA) in parallel with 70 μL of Droplet Generation Oil for Probes (Bio-Rad, Hercules, CA, USA) and placed on the QX200™ Droplet Generator (Bio-Rad, Hercules, CA, USA). Forty microliters of droplets were then dispensed to a Multiplate® PCR Plate™ (Low 96-well Clear) (Bio-Rad, Hercules, CA, USA), sealed with Pierceable Foil Heat Seal (Bio-Rad, Hercules, CA, USA) in the PX1™ PCR plate sealer (Bio-Rad, Hercules, CA, USA), and the plate was placed on the T100™ Thermal Cycler (Bio-Rad, Hercules, CA, USA) for the PCR reaction as follows: enzyme activation at 95 °C for 10 min followed by 40 cycles of denaturation at 94 °C for 30 s and annealing/extension at 55 °C for 2 min. Following PCR amplification, the droplets were read using the QX200™ Droplet Reader (Bio-Rad, Hercules, CA, USA), and through the QuantaSoft™ software the positive and negative droplets were counted based on the emitted fluorescence intensity.

**3. Results**

The key aspect of droplet digital PCR, its ability to provide single molecule analysis, was exploited for the direct elucidation of haplotypes and chromosomal phasing in the parents undergoing prenatal diagnosis. This is performed by genotyping the parental genomic DNA samples exclusively, thus bypassing the traditional necessity for analyzing other family members' genetic material.

To prepare this assay, four highly-heterozygous SNVs determined from the study of Papasavva et al. [17] [SNV rs1003586 (G/A), SNV rs11036364 (T/C), SNV rs7480526 (G/T), SNV rs2187610 (G/C)], and the pathogenic SNV IVSI-110 (c.93-21G>A) were initially selected to be analyzed. The SNVs were selected such that two were located upstream of the pathogenic variant [SNV rs1003586 (G/A) and SNV rs11036364 (T/C)], whereas the other two were located downstream of the pathogenic variant [SNV rs7480526 (G/T) and the SNV rs2187610 (G/C)], to prevent false results due to possible recombination events.

The efforts were focused on the analysis of nine family trios (mother, father, and CVS). For each trio, 36 reactions were performed in total. Ten duplex reactions were performed for each sample of genomic DNA, following the duplexing procedure described in Table 4 and Figure 3. Moreover, five reactions correspond to the heterozygous controls (one for each targeted SNV), while one reaction without any template was also performed to prevent possible false positive results due to contamination events.

As mentioned earlier, our chromosomal phasing approach is based on the hypothesis that when two SNV alleles are linked and co-inherited, they are expected to partition into

the same droplet, greatly surpassing chance expectations, thus emitting a higher double positive signal than trans-configured alleles.

An example of the trio analysis of family seven is presented in Figures 4–6.

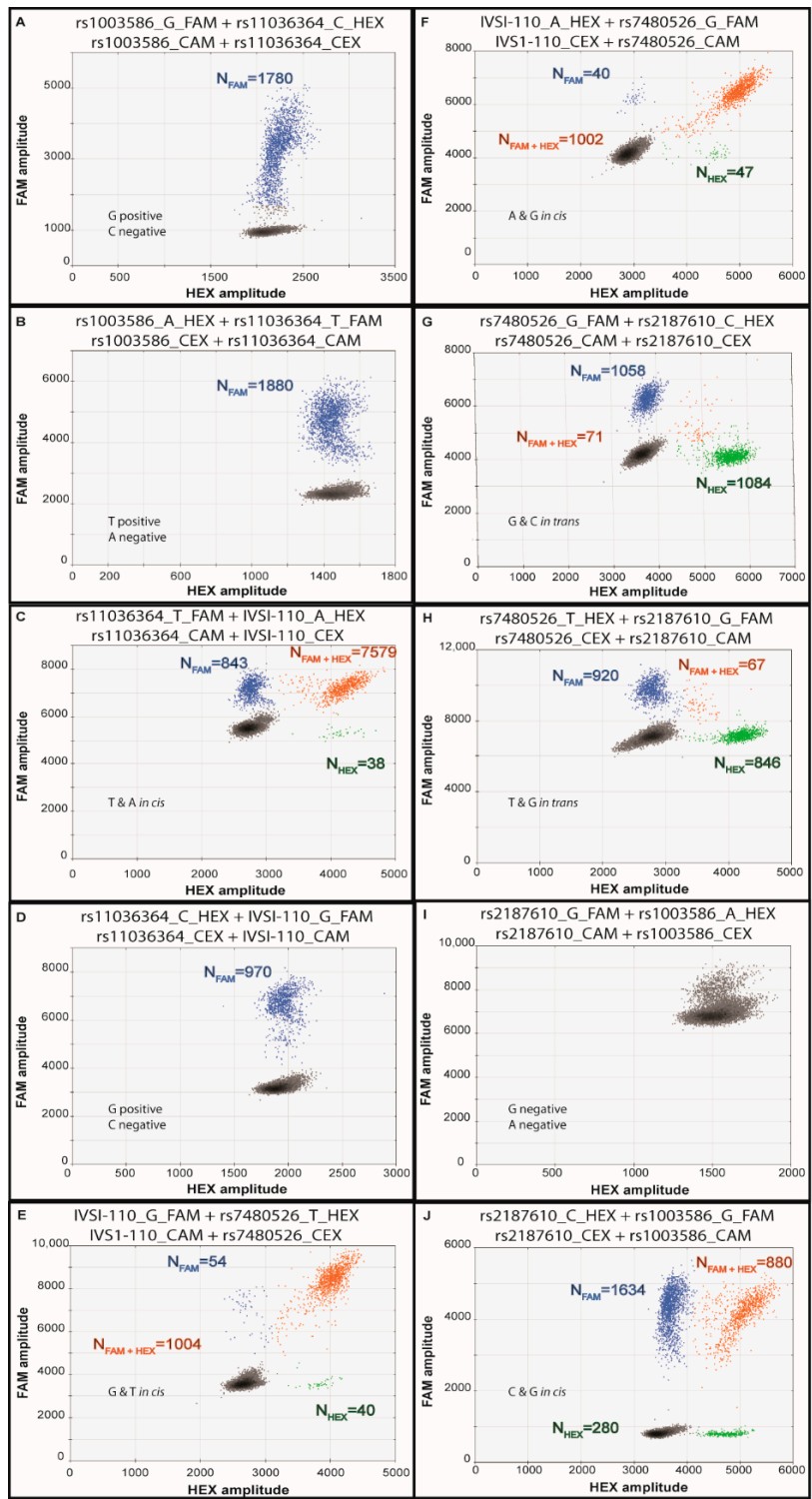

**Figure 4.** Chromosomal phasing for the mother of family 7. Blue: FAM, Green: HEX, Orange: positive for both fluorophores (co-partitioning alleles), Black: negative (absence of fluorophores), N = number of droplets, CAM: Competitor for FAM, and CEX: Competitor for HEX. (**A**–**J**): 2D-plots of each of the ten duplex reactions that have been performed for the direct chromosomal phasing for the mother of family 7. The duplexed probes are presented on top of each subfigure.

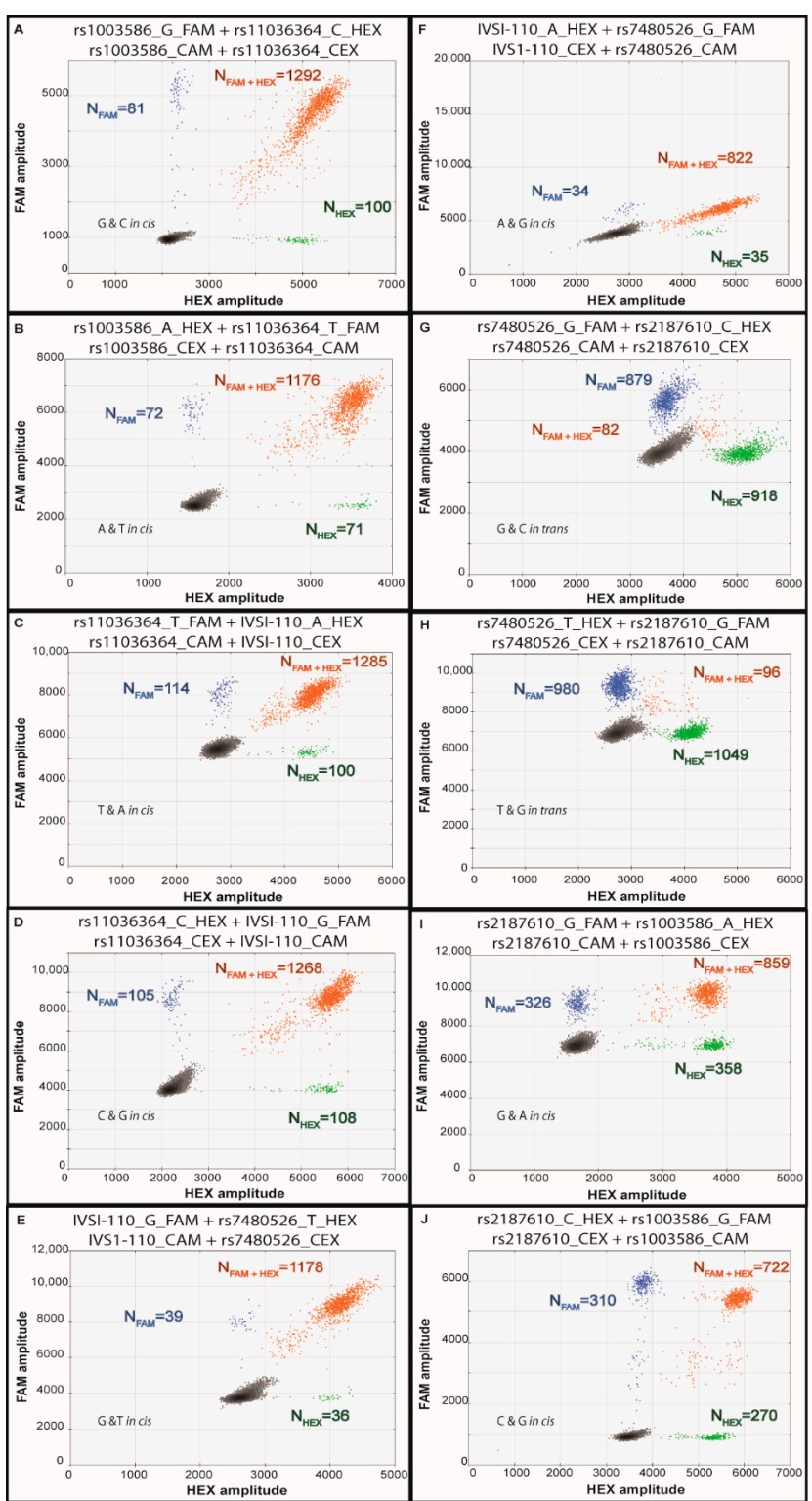

**Figure 5.** Chromosomal phasing for the father of family 7. Blue: FAM, Green: HEX, Orange: positive for both fluorophores (co-partitioning alleles), Black: negative (absence of fluorophores), N = number of droplets, CAM: Competitor for FAM, and CEX: Competitor for HEX. (**A**–**J**): 2D-plots of each of the ten duplex reactions that have been performed for the direct chromosomal phasing for the father of family 7. The duplexed probes are presented on top of each subfigure.

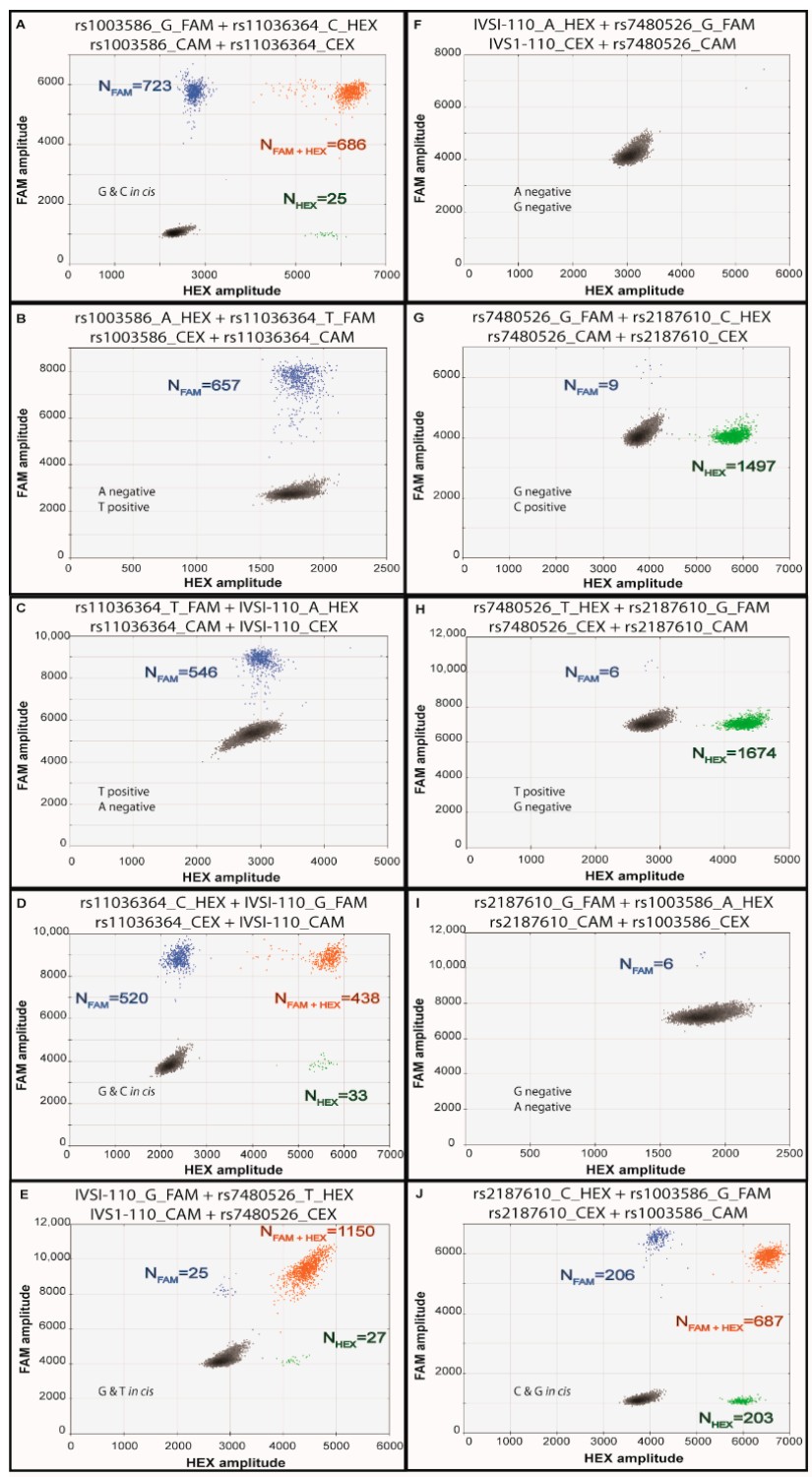

**Figure 6.** Chromosomal phasing for the fetus of family 7. Blue: FAM, Green: HEX, Orange: positive for both fluorophores (co-partitioning alleles), Black: negative (absence of fluorophores), N = number of droplets, CAM: Competitor for FAM, and CEX: Competitor for HEX. (**A**–**J**): 2D-plots of each of the ten duplex reactions that have been performed for the direct chromosomal phasing for the fetus of family 7. The duplexed probes are presented on top of each subfigure.

Figure 4 demonstrates the 10 reactions performed for the analysis of the maternal genome. In Figure 4A, the FAM probe that targets the G allele of SNV rs1003586 is duplexed with the HEX probe that targets the C allele of SNV rs11036364. The emission of only the FAM fluorophore indicates that the mother is positive for the G allele of SNV rs1003586 and negative for the C allele of SNV rs11036364. Using HEX to target the A allele of SNV rs1003586 and FAM to target the T allele of SNV rs11036364 results in the emission of solely the FAM fluorophore, designating that the mother is positive for the T allele of SNV rs11036364 and negative for the A allele of SNV rs1003586 (Figure 4B). Hence, the conclusion was drawn that the mother is homozygous G for the SNV rs1003586 and homozygous T for the SNV rs11036364. Next, Figure 4C illustrates that the T allele of SNV rs11036364 (FAM) results in a high number of double positive droplets when it is duplexed with the A allele of the pathogenic SNV IVSI-110 (c.93-21G>A), providing an indication that the mother carries the β-thal allele. When the G allele of the pathogenic SNV IVSI-110 (c.93-21G>A) (FAM) was duplexed with the C allele of the SNV rs11036364 (HEX), only the FAM fluorophore emits a signal (Figure 4D), proving the carrier status of the mother for the pathogenic SNV IVSI-110 (c.93-21G>A) and confirming the absence of the C allele of the SNV rs11036364, as previously determined. Furthermore, the G allele of the pathogenic SNV IVSI-110 (c.93-21G>A) (FAM) was duplexed with the T allele of the SNV rs7480526 (HEX), resulting in the high number of double positive events (Figure 4E). Hence, a suspicion arose that these two targeted alleles are located on the same chromosome. For further evidence, when the A allele of the pathogenic SNV IVSI-110 (c.93-21G>A) (HEX) was duplexed with the G allele of the SNV rs7480526 (FAM), a high double positive signal indicated their co-inheritance (Figure 4F). Moreover, the low number of double positive events in Figure 4G when targeting the G allele of the SNV rs7480526 (FAM) and the C allele of the SNV rs2187610 (HEX) pinpoints their localization on different chromosomal molecules. To this extent, the duplexing of the T allele of the SNV rs7480526 (HEX) with the G allele of the SNV rs2187610 (FAM) presented in Figure 4H specifies their trans-configuration due to the low double positive signal. Next, the G allele of the SNV rs2187610 (FAM) was duplexed with the A allele of the SNV rs1003586 (HEX). The double negative signal shown in Figure 4I was a false negative result since the mother is positive for the G allele of the SNV rs2187610, and thus FAM positive droplets were expected. In addition, when the C allele of the SNV rs2187610 (HEX) was duplexed with the G allele of the SNV rs1003586 (FAM), true positive signals were detected, proving the presence of both alleles in the maternal genome (Figure 4J).

The reactions performed for the chromosomal phasing of the father are illustrated in Figure 5. In Figure 5A, the allele G of the SNV rs1003586 (FAM) is duplexed with the allele C of the SNV rs11036364 (HEX), and since the number of double positive droplets greatly exceeds chance expectations, an assumption was made that the two alleles are cis-configured. To this extent, when we targeted the A allele of the SNV rs1003586 (HEX) and the allele T of the SNV rs11036364 (FAM) (Figure 5B), their co-inheritance was determined due to a high double positive signal. Furthermore, the T allele of the SNV rs11036364 (FAM) was duplexed with the A allele of the pathogenic SNV IVSI-110 (c.93-21G>A) (HEX), and the high number of double positive events pointed towards their cis-configuration (Figure 5C). Furthermore, a high double positive signal was also emitted when the C allele of the SNV rs11036364 (HEX) was duplexed with the G allele of the pathogenic SNV IVSI-110 (c.93-21G>A) (FAM), designating their localization on the same chromosomal molecules (Figure 5D). In addition, when the G allele of the pathogenic SNV IVSI-110 (c.93-21G>A) (FAM) was duplexed with the T allele of the SNV rs7480526 (HEX), their coinheritance became evident due to the high level of double positive events (Figure 5E) and was cross-confirmed when the allele A of the pathogenic SNV IVSI-110 (c.93-21G>A) (HEX) and the allele G of the SNV rs7480526 (FAM) were duplexed and also proved their cis-configuration (Figure 5F). The trans-configuration of the allele G of the SNV rs7480526 (FAM) with the C allele of the SNV rs2187610 (HEX) became evident when their duplexing resulted in a low double positive signal (Figure 5G). To this extent, the T allele of the SNV rs7480526

(HEX) and the G allele of the SNV rs2187610 (FAM) were multiplexed and specified their localization on separated chromosomal molecules (Figure 5H). In Figure 5I, the FAM probe that targets the G allele of SNV rs2187610 is duplexed with the HEX probe that targets the A allele of SNV rs1003586. The emission of a high double positive signal confirmed that the two alleles were linked. Using HEX to target the C allele of SNV rs2187610 and FAM to target the G allele of SNV rs1003586 again resulted in a high number of double positive events (Figure 5J), which verified that the two alleles were located in *cis*.

Figure 6 presents the 10 multiplexed reactions performed on the CVS sample for the direct chromosomal phasing of the fetus. The prominent signal of the double positive events in Figure 6A when the G allele of SNV rs1003586 (FAM) is duplexed with the C allele of SNV rs11036364 (HEX) designates their cis-configuration. However, the emission of only the FAM fluorophore when the A allele of SNV rs1003586 (HEX) was multiplexed with the T allele of SNV rs11036364 (FAM) points out the homozygosity of the fetus for the G allele of SNV rs1003586 (Figure 6B). Moreover, the absence of the β-thal allele A of the pathogenic SNV IVSI-110 (c.93-21G>A) was determined when its HEX-labeled probe was duplexed with the FAM-labeled probe targeting the T allele of SNV rs11036364 (FAM), whereby the emission of only the FAM signal was observed (Figure 6C). In addition, when the C allele of the SNV rs11036364 (HEX) was simultaneously targeted with the G allele of the pathogenic SNV IVSI-110 (c.93-21G>A) (FAM), true positive signals for both targeted alleles were observed (Figure 6D), confirming the homozygosity of the fetus for the wild-type G allele of the pathogenic SNV IVSI-110 (c.93-21G>A) and the presence of the C allele for SNV rs11036364. At this point, the phase of SNV rs11036364 could not be determined since it was duplexed with two homozygous SNVs. Moreover, the G allele of the pathogenic SNV IVSI-110 (c.93-21G>A) (FAM) was duplexed with the T allele of the SNV rs7480526 (HEX). The double positive signal's prominence would indicate the coinheritance of the two targeted alleles if the fetus was not homozygous for the G allele of the pathogenic SNV IVSI-110 (c.93-21G>A) (Figure 6E). Furthermore, the negative signal illustrated in Figure 6F represents the absence of both targeted alleles from the fetal genome, i.e., the allele A of the pathogenic SNV IVSI-110 (c.93-21G>A) and the allele G of the SNV rs7480526. Confirming the homozygosity of the T allele of SNV rs7480526, the multiplexing of the G allele (FAM) of the aforementioned SNV with the C allele of the SNV rs2187610 (HEX) resulted in no emission of the FAM fluorophore (Figure 6G), whereas the duplexing of the T allele of SNV rs7480526 (HEX) with the G allele of the SNV rs2187610 (FAM) yielded only the emission of the HEX signal (Figure 6H). To this extent, the absence of the allele G and the homozygosity of the C allele of SNV rs2187610 are suspected. When the G allele of the SNV rs2187610 (FAM) was duplexed with the A allele of the SNV rs1003586 (HEX) (Figure 6I), the homozygosity of the C allele of SNV rs2187610 and the G allele of SNV rs1003586 were indirectly confirmed since a negative signal was emitted. The simultaneous targeting of the C allele of SNV rs2187610 (HEX) with the G allele of SNV rs1003586 (FAM), presented in Figure 6J, resulted in a high double positive signal emission and the direct confirmation of the homozygosity of both targeted SNV alleles.

A DNA sample from an individual who is heterozygous for all five of the SNVs tested was used as a control. As expected, the low signal of double positive events was detected in the five reactions performed since heterozygous alleles are located by default on different chromosomal molecules (Figure 7A–E). A no template control was also included in the experiment with randomly selected primers and probes for duplexing, which yielded no positive events for either fluorophore, as expected (Figure 7).

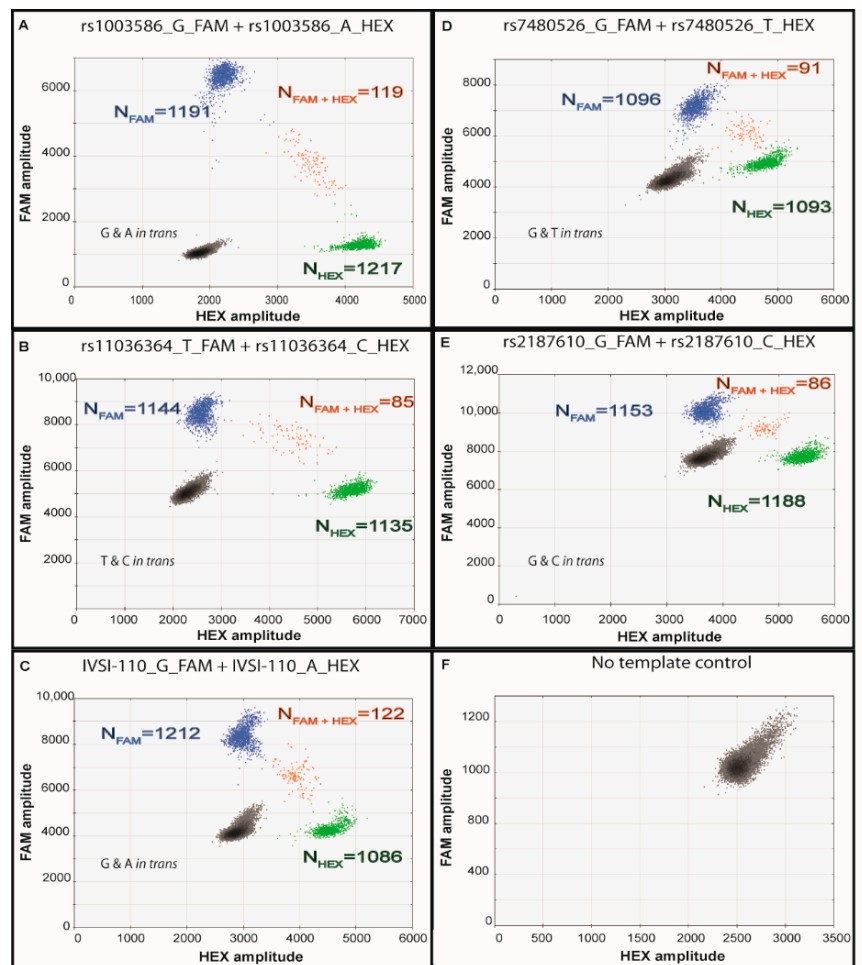

**Figure 7.** Controls for chromosomal phasing of family 7. (**A**) The G allele (FAM) of SNV rs1003586 is duplexed with the A allele (HEX) of the same SNV. (**B**) The T allele (FAM) of the SNV rs11036364 is duplexed with the C allele (HEX) of the same SNV. (**C**) The G allele (FAM) of the pathogenic variant IVSI-110 (c.93-21G>A) is duplexed with the A allele of the same variant. (**D**) The G allele (FAM) of the SNV rs7480526 is duplexed with the T allele (HEX) of the same SNV. (**E**) The G allele (FAM) of the SNV rs2187610 is duplexed with the C allele (HEX) of the same SNV. (**F**) No template control. Blue: FAM, Green: HEX, Orange: positive for both fluorophores (co-partitioning alleles), Black: negative (absence of fluorophores), and N = number of droplets.

Relying on the above results, the successful chromosomal phasing for the three samples was achieved and resulted in the direct chromosomal phasing and haplotyping of family seven. Following the same procedure, the analysis of the remaining eight family trios was performed, resulting in the successful chromosomal phasing and haplotyping of all nine of the families analyzed. For the proof of principle, the direct phasing outcome of ddPCR was compared, for each family, with the haplotypes ascertained through traditional methods based on family trios, demonstrating full concordance in all cases (Figure 8). At the same time, our presented assay accomplishes chromosomal phasing with simultaneous SNV genotyping that also proved to agree with previous MALDI-TOF MS genotyping analyses (Table 5).

In the event where an SNV was duplexed with two homozygous SNVs (green) (Figure 8), a direct phasing was not possible. However, the phase could still be determined based on genotypic information obtained from the ddPCR analysis of the family trio. It is important to note that the phase of SNV rs11036364 for the parents and the fetus of family nine could not be determined using the conventional family trio phasing approach since all three samples were heterozygous. Our approach for direct phasing using ddPCR

is capable of determining the phase of this SNV, overcoming the limitation inherent in traditional phasing that is based on family studies for heterozygous SNVs shared by all family members (Figure 8).

**Figure 8.** Direct chromosomal phasing and haplotyping for 9 families using ddPCR (**bottom**) and comparison with family trio phasing (**top**). The β-thalassaemia allele is marked with red. Variants for which the phase was not able to be determined are shown with green. The fetus has inherited the wild-type haplotype from both parents in families 3, 4, 6, 7, and 9. In contrast, the fetus in families 1, 2, 5, and 8 has inherited one β-thalassaemia allele from one parent. CVS: Chorionic Villi Sample, β-thal: β-thalassaemia haplotype, WT: Wild-type haplotype, $P_{WT}$: Paternal wild-type, $M_{WT}$: Maternal wild-type, $P_{\beta\text{-thal}}$: Paternal β-thalassaemia, and $M_{\beta\text{-thal}}$: Maternal β-thalassaemia.

**Table 5.** SNV genotyping results of MALDI-TOF MS for all the samples analyzed.

| Family No. | | rs1003586 (G/A) | rs11036364 (T/C) | IVSI-110 (c.93-21G>A) | rs7480526 (G/T) | rs2187610 (G/C) |
|---|---|---|---|---|---|---|
| 1 | mother | G | CT | G | T | C |
| | father | GA | CT | GA | GT | GC |
| | CVS | GA | T | GA | GT | GC |
| 2 | mother | GA | CT | G | GT | GC |
| | father | GA | CT | GA | GT | GC |
| | CVS | A | T | GA | G | G |
| 3 | mother | GA | T | GA | GT | GC |
| | father | GA | CT | GA | G | G |
| | CVS | G | CT | G | GT | GC |
| 4 | mother | GA | CT | GA | GT | GC |
| | father | GA | T | GA | G | G |
| | CVS | G | CT | G | GT | GC |

**Table 5.** *Cont.*

| Family No. | | rs1003586 (G/A) | rs11036364 (T/C) | IVSI-110 (c.93-21G>A) | rs7480526 (G/T) | rs2187610 (G/C) |
|---|---|---|---|---|---|---|
| 5 | mother | G | CT | G | T | C |
| | father | GA | T | GA | GT | GC |
| | CVS | GA | CT | GA | GT | GC |
| 6 | mother | A | T | GA | GT | GC |
| | father | GA | CT | GA | GT | GC |
| | CVS | GA | CT | G | T | C |
| 7 | mother | G | T | GA | GT | GC |
| | father | GA | CT | GA | GT | GC |
| | CVS | G | CT | G | T | C |
| 8 | mother | GA | T | GA | GT | GC |
| | father | G | CT | G | G | GC |
| | CVS | GA | CT | GA | G | G |
| 9 | mother | GA | CT | GA | GT | GC |
| | father | G | CT | GA | GT | GC |
| | CVS | G | CT | G | T | C |

Significantly, we have presented our developed assay and its feasibility on the construction of haplotypic blocks and the determination of the phase of the fetus's inherited alleles, with the ultimate goal of its introduction to β-thalassaemia prenatal diagnostics. More specifically, we have shown that in five out of the nine families examined (families three, four, six, seven, and nine), the fetus has inherited both wild-type alleles from its parents while in the remaining four families (family one, two, five, and eight) the fetus is a β-thalassaemia carrier. In addition, and importantly, the fetal carrier status results obtained from the ddPCR assay were confirmed with direct mutation detection analysis (i.e., Real-time PCR), again showing full concordance in all the families tested (Table 6).

**Table 6.** Confirmation of the fetal carrier status ddPCR-based results with real-time PCR (direct mutation detection).

| Family No. | Fetal Carrier Status Based on ddPCR | Fetal Carrier Status Based on Real-Time PCR (Direct Mutation Detection) | Concordance |
|---|---|---|---|
| 1 | β-thalassaemia carrier | β-thalassaemia carrier | Yes |
| 2 | β-thalassaemia carrier | β-thalassaemia carrier | Yes |
| 3 | Absence of the β-thalassaemia pathogenic haplotype | Absence of the β-thalassaemia pathogenic allele | Yes |
| 4 | Absence of the β-thalassaemia pathogenic haplotype | Absence of the β-thalassaemia pathogenic allele | Yes |
| 5 | β-thalassaemia carrier | β-thalassaemia carrier | Yes |
| 6 | Absence of the β-thalassaemia pathogenic haplotype | Absence of the β-thalassaemia pathogenic allele | Yes |
| 7 | Absence of the β-thalassaemia pathogenic haplotype | Absence of the β-thalassaemia pathogenic allele | Yes |
| 8 | β-thalassaemia carrier | β-thalassaemia carrier | Yes |
| 9 | Absence of the β-thalassaemia pathogenic haplotype | Absence of the β-thalassaemia pathogenic allele | Yes |

## 4. Discussion

Haplotype phasing strategies have been traditionally dependent on the analysis of related family members (grandparents or siblings), thereby restricting their application to families for which such samples are available. In this study, we took advantage of the single molecule analysis potentials of ddPCR and demonstrated that the haplotypic arrangements of an individual can be resolved directly, without the need to rely on family studies. A direct chromosomal phasing assay for the prenatal diagnosis of β-thalassaemia has been developed by our group and applied to nine family trios at risk of having a β-thalassaemia-affected child for the proof of concept. By using five β-globin gene cluster variants, the construction of the haplotypic blocks was performed and the ascertainment of the allelic phase was achieved in all of the families. In this study, we have proved that our direct chromosomal phasing approach can facilitate haplotype analysis, and along with a direct mutation detection method, can be used in prenatal diagnostics, while simultaneously providing a great opportunity to families with no other family members available for testing.

The aforementioned assay holds many advantages as it is highly accurate, easy to implement, time-efficient, and inexpensive. Significantly, the fact that the need for analyzing other family members' genetic material is bypassed sets this approach in the forefront, especially in cases of dispersed populations. However, a limitation was observed for homozygous SNVs that do not provide information about the phase of the allele. Therefore, the success and the reliability of the haplotype phasing depends on the use of heterozygous SNVs. To overcome this limitation and obtain allelic phase information, the identification and multiplexing of heterozygous SNVs should be performed. However, this would require genotyping a priori, which is time and cost inefficient. Another way to bypass the restriction for allelic phase information caused by the homozygous variants is to include a large number of SNVs in the assay. Hence, the likelihood of having more heterozygous variants, and enough for a reliable analysis, is increased.

One false negative result was obtained during the mother's analysis in family seven where the G allele of rs2187610 was duplexed with the A allele of rs1003586. This phenomenon might be attributed to a pipetting error since the whole experimental procedure was performed manually. This should be overcome by performing a series of automated procedures, from sample injection to analysis, thus minimizing the risks inherent to manual handling [18]. It is important to emphasize that despite the false negative result obtained, the correct chromosomal phasing and haplotyping was achieved.

Our approach, which is based on the duplexing of each allele of one SNV loci with two other interchanging SNV loci, proved to be successful in ascertaining the chromosomal phase in all the families analyzed. The multiple outcomes that this approach has regarding each SNV loci work synergistically to the final inference of phasing by cross-confirming the results and minimizing the risk for false interpretations.

Others have obtained direct chromosomal phasing based on other successful approaches such as linked-read sequencing [19] and the TLA [20] method. However, the linked-read technology is based on specific instruments such as 10× Genomics that may not be available in every clinical laboratory. On the other hand, due to the broad range of applications provided by a ddPCR system, it has recently been introduced into many laboratories. Nonetheless, while the TLA is a sensitive and effective approach, it requires additional laboratory procedures to phase the alleles since it requires cells as an input material, thus contributing to a procedure that is laborious and time inefficient, especially in the diagnostic setting where the turnaround time of delivering results is of essence.

Our approach proved to be robust, cost- and time-effective, and easy to implement and analyze. As the family studies require the analysis of additional family members such as grandparents or siblings who are homozygote wild-type or homozygote mutant, this approach can provide benefits for couples even on their first pregnancy or whose parents are not available at the time of testing, especially in countries with a high population density. Nevertheless, a large-scale analysis with the inclusion of a large number of SNVs

should be implemented in near future in order to broaden the applicability of the method and minimize the risk of error. Moreover, a validation analysis via an examination of more families should also be performed to ascertain the validity and accuracy of the method. The validated method could potentially be implemented in routine prenatal diagnosis in cases where family members are not available and therefore prenatal diagnosis based on haplotyping cannot be performed. The utility of the direct haplotype phasing method presented above will also provide a model for the development of prenatal diagnostic assays for other monogenic inheritable diseases. Apart from this, this approach can also benefit other applications, such as allele-specific expression studies, the identification of compound heterozygosity, and genome engineering.

**Author Contributions:** Conceptualization, S.B., M.K. and T.P.; Data curation, S.B., M.K. and T.P.; Formal analysis, S.B. and T.P.; Funding acquisition, M.K. and T.P.; Investigation, S.B.; Methodology, S.B., G.C. and T.P.; Project administration, M.K. and T.P.; Resources, A.C., C.M. and C.I.; Software, S.B.; Supervision, M.K. and T.P.; Validation, S.B.; Visualization, S.B.; Writing—original draft, S.B. and T.P.; Writing—review & editing, G.C., A.C., C.M., C.I., M.K. and T.P. All authors have read and agreed to the published version of the manuscript.

**Funding:** This research was funded by the Excellence Hubs Program of The Research and Innovation Foundation (RIF), "Restart 2016–2020", Excellence/1216/0484.

**Institutional Review Board Statement:** The study was conducted in accordance with the Declaration of Helsinki and approved by the Cyprus National Bioethics Committee (EEBK/ΕΠ/2018/51) on 7 February 2019.

**Informed Consent Statement:** Informed consent was obtained from all subjects involved in the study.

**Data Availability Statement:** Not applicable.

**Acknowledgments:** We would like to thank Aekaterini Mavri for her artistic input in Figure 3.

**Conflicts of Interest:** The authors declare no conflict of interest. The funders had no role in the design of the study; in the collection, analyses, or interpretation of data; in the writing of the manuscript, or in the decision to publish the results.

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
