# Peer review of "Direct Chromosomal Phasing: An Easy and Fast Approach for Broadening Prenatal Diagnostic Applicability"

_thalassrep, doi:10.3390/thalassrep12030011_

Round 1

Reviewer 1 Report

The authors present an approach using Droplet Digital PCR to directly determine haplotypes using four highly polymorphic SNPs, overcoming the necessity for other family members. They consider that the clinical utility of this approach can open up the application of prenatal diagnosis for β-thalassemia and other monogenic disorders.

 This is an interesting method to determine haplotypes in individual samples without needing of relative’s DNA samples however, it is not clear to this reviewer how this can be important for prenatal diagnosis of beta-thalassemia or other monogenic diseases. Therefore, the significance and utility of the direct haplotype phasing method for prenatal diagnosis should be clearly explained in the manuscript.

Minor point:

 The last sentence in Results section (pg.20 L.691.693) should be moved to Discussion.

Reviewer 2 Report

This is an interesting paper that shows the application of ddPCR, a methodology whose use is increasing in many laboratories, to determine the chromosomal phasing of haplotypes in prenatal diagnosis. 

The manuscript is well-written and gives enough reasons of its potential relevance. It has demonstrated that this methodology could be implemented in the future for rutinary prenatal diagnosis, when stablished methods cannot be used.

My only comment is for figure 2, which I could not see for being absent from the pdf. 

Round 2

Reviewer 1 Report

I have no further comments on this manuscript.